# Performance Analysis and Architecture of a Clustering Hybrid Algorithm Called FA+GA-DBSCAN Using Artificial Datasets

**DOI:** 10.3390/e24070875

**Published:** 2022-06-25

**Authors:** Juan Carlos Perafan-Lopez, Valeria Lucía Ferrer-Gregory, César Nieto-Londoño, Julián Sierra-Pérez

**Affiliations:** 1Grupo de Investigación en Ingeniería Aeroespacial, Universidad Pontificia Bolivariana, Medellín 050031, Colombia; julian.sierra@upb.edu.co; 2Semillero de Investigación en Ingeniería Aeroespacial, Universidad Pontificia Bolivariana, Medellín 050031, Colombia; valeria.ferrer@upb.edu.co; 3Grupo de Investigación en Energía y Termodinámica, Universidad Pontificia Bolivariana, Medellín 050031, Colombia; cesar.nieto@upb.edu.co

**Keywords:** clustering, DBSCAN, factor analysis, genetic algorithm, pattern recognition, entropy

## Abstract

Density-Based Spatial Clustering of Applications with Noise (DBSCAN) is a widely used algorithm for exploratory clustering applications. Despite the DBSCAN algorithm being considered an unsupervised pattern recognition method, it has two parameters that must be tuned prior to the clustering process in order to reduce uncertainties, the minimum number of points in a clustering segmentation MinPts, and the radii around selected points from a specific dataset Eps. This article presents the performance of a clustering hybrid algorithm for automatically grouping datasets into a two-dimensional space using the well-known algorithm DBSCAN. Here, the function nearest neighbor and a genetic algorithm were used for the automation of parameters MinPts and Eps. Furthermore, the Factor Analysis (FA) method was defined for pre-processing through a dimensionality reduction of high-dimensional datasets with dimensions greater than two. Finally, the performance of the clustering algorithm called FA+GA-DBSCAN was evaluated using artificial datasets. In addition, the precision and Entropy of the clustering hybrid algorithm were measured, which showed there was less probability of error in clustering the most condensed datasets.

## 1. Introduction

Data classification has recently become an essential activity in solving and handling problems in which large datasets are involved. When used as a tool for data classification, pattern recognition algorithms may have a fundamental implication in the decision-making process. There are two prominent methods defined for data classification: supervised classification, where the number of groups has to be defined before classification, and unsupervised classification, in which it is expected that the algorithm performs the clustering analysis by itself without requiring a previous setup of the parameters. Density clustering techniques are a subgroup of unsupervised pattern recognition algorithms. These algorithms involve methodologies to identify a particular density in the space of points from a specific dataset. Therefore, density interrelated elements will be included in a particular cluster. As a result, density-based clustering algorithms can determine clusters with a significant diversity of shapes and discriminate meaningful information from outliers [1,2].

Density-Based Spatial Clustering of Application with Noise (DBSCAN) is a density-based unsupervised classification method developed by Easter et al. and presented in 1996 [3]; however, it is still recognized as a helpful method due to its simplicity and good overall performance [4]. Density-based algorithms such as DBSCAN, Clustering Large Applications based on Randomized Search (CLARANS) [5], and Ordering Points To Identify Clustering Structure (OPTICS) [6], among others [7,8], are used for exploratory analysis, where classes belonging to a specific dataset *D* are not completely defined or the set is by nature randomized. Furthermore, unlike popular clustering algorithms, such as K-means, DBSCAN does not require limiting the number of clusters or classes previously.

The DBSCAN algorithm is not entirely automatized, and defining two input parameters that depend on the analyzed dataset *D* is thus needed to perform the clustering process. These input parameters are Eps and MinPts, which depend on the density and magnitude of the specific dataset *D* being examined. Eps is the radius of an imaginary circumference where a minimum set of points is reachable when using the Euclidean distance defined by the parameter MinPts. Although DBSCAN was first introduced in 1996, the algorithm is still widely used today, being awarded the SIGKDD Test-of-Time Award in 2014. Its relevance is associated with its ease of implementation and reasonable computational cost, $O(n^{3})$, when used in large datasets [4]. Nevertheless, the algorithm may achieve an acceptable computational cost, as well as its precision will vastly depend on the selection of parameters Eps and MinPts, since these parameters must be adjusted according to the specific dataset being analyzed.

The DBSCAN algorithm measures distance from point to point using the well-known Euclidean distance, i.e., DBSCAN could perform a clustering analysis in dimensional spaces greater than two. Even though the Euclidean distance can be measured in an *n*-dimensional space, the clustering process entails a lower computational complexity cost when performed in a two-dimensional space. Consequently, if the dataset being analyzed has a dimension higher than two, it is deemed necessary to evaluate the use of a strategy for dimensional reduction as part of pre-processing, such as Principal Component Analysis (PCA) or the Factor Analysis (FA) method. Datasets can be handled before clustering using a dimensionality reduction method. Although a fraction of information is lost, benefits can be expected in the overall clustering process by reducing computational costs.

Many variations of the DBSCAN algorithm have been developed to obtain a highly autonomous and precise algorithm with the lowest possible computational cost. In addition, different performance metrics can be used as algorithm evaluation metrics to improve their performance. BIRCHSCAN is an algorithm presented by de Moura [9], where the BIRCH algorithm was merged with DBSCAN as a strategy for significant dataset clustering. The CF-Three method and a threshold were determined for the Eps parameter selection. The BIRCH algorithm is defined to evaluate the dataset to select a smaller representative biased sub-dataset, which is evaluated using DBSCAN. The evaluation metrics selected for this methodology are the Rand Index and the Adjusted Rand Index.

Lai et al. [10] presented a method based on Multi-Verse Optimization (MVO) to improve the selection of DBSCAN parameters Eps and MinPts using the *r* rates in the Accuracy of artificial datasets. In the study proposed by Wang et al. [11], a method for automatic estimation of the DBSCAN parameter Eps was defined for LiDAR data segmentation clustering. The estimation of the parameter Eps was based on the average value in the population defined by the nearest neighbor function. The accuracy of the results was estimated using reference data.

The paper presented by Darong and Peng [12] combined a grid partition technique with DBSCAN, calling the methodology GRPDBSCAN. The strategy implies partitioning the information on grids and then finding the suitable DBSCAN parameters considering the information contained in each partition. Although the authors emphasized the algorithm’s precision, it was not clear how the clustering performance was measured in this work. Ohadi et al. defined a new DBSCAN algorithm called SW-DBSCAN [13] formulated on the sliding window grid-based model [14]. Nevertheless, in this paper, the evaluation of the algorithm was measured using the Accuracy metric. The algorithm BDE-DBSCAN proposed by Karami and Johansson [15] presents a methodology for automatic DBSCAN parameter definition using a hybrid optimization method called Binary Differential Evolution. An analytical process and the Tournament Selection (TS) technique were selected for Eps estimation. The performance of the algorithm was defined using the effectiveness metric.

The work presented by Kumar and Reddy [16] adopted a methodology based on structures associated with specific groups that accelerate the neighborhood search queries. As a result, the clustering technique increased DBSCAN’s clustering performance by 2.2. This method is called G-DBSCAN, an accelerated DBSCAN algorithm that aims to find the nearest neighbor with the help of group methods. In short, the algorithm works by applying grouping partition methods to identify subgroups with similar patterns in a specific dataset *D*, which is followed by a dimensional reduction method and the definition of the parameter Eps for each group.

Zhu et al. [17] defined a methodology for an adaptive Eps parameter estimation implementing a Gauss kernel density method considering the clustering of unbalanced artificial datasets. Clustering performance was evaluated using the Rand Index and V-measure. A novel algorithm was presented in [18]; this algorithm, called K-DBSCAN, is considered as an optimization algorithm called Harmony Search (HS), which designates the proper value of the clustering parameters. Here, the cluster number K is predefined by a partition clustering approach. The HS algorithm defines the optimal value for the DBSCAN parameters. The Rand Index and Jaccard coefficient are the evaluation metrics selected to measure the algorithm’s effectiveness.

A parameter-free method called Dsets-DBSCAN was reported by Hou et al. [19]. A histogram equalization transformation of similarity matrices was executed in this work to create a dominant set of independent parameters. The quality of the results was estimated using the F-measure metric. The results showed a remarkable performance of the parameter-free algorithm. The methodology presented in [20,21] also used a parameter-free clustering process for DBSCAN using the nearest neighbor function commonly denoted as *k*-dist. The evaluation of the algorithm was performed by visual inspection of the results. Ozkok and Celik [22] presented a novel algorithm called AE-DBSCAN, which included a method for the automatic definition of parameters Eps and MinPts. They also considered the *k*-dist by using the nearest neighbor function. Soni and Ganatra [23] proposed a new algorithm called AGED. The methodology defines a group of densities extracted from the dataset clustered using the well-known nearest neighbor function, specifically the *k*-dist plot. This work evaluated a variety of performance metrics, including the Dunn Index, the Pearson Gamma coefficient, and the Entropy. Other methodologies based on the DBSCAN algorithm were presented in [24,25,26,27,28].

As shown above, a large variety of metrics have been employed. However, Entropy as a metric has not been quite used for the performance evaluation of the DBSCAN algorithm and its variants. Here lies the intention of using Entropy as an evaluation metric considering the information given by DBSCAN after performing the clustering analysis. Entropy will manifest the orderly clustering in which results with values close to 0 are considered and grouped into datasets. In this work, the performance of a clustering hybrid algorithm called FA+GA-DBSCAN is presented, taking into account the DBSCAN algorithm as its core. This unsupervised pattern recognition algorithm was at first developed to identify the operational conditions in a structure under a variety of loads [29]. Moreover, in order to define adequate values for Eps and MinPts, a Genetic Algorithm (GA) was implemented. The GA was based on a randomized population extracted from a particular dataset *D* using distances selected by the nearest neighbor function and also included a set of points (x,y) belonging to the dataset *D* being examined. Later, a radius that represents the parameter Eps was found. In this work, the data preprocessing, including normalization and data reduction, is shown in Section 2. The definition of the DBSCAN parameters is specified in Section 3. The evaluation of FA+GA-DBSCAN is performed in Section 4. Two case studies using FA+GA-DBSCAN are presented in Section 5. Conclusions are made in Section 6.

## 2. Data Preprocessing

A large amount of information is collected from experiments related to knowledge discovery problems. Therefore, it is expected that under a non-trivial process, novel and potentially useful information is extracted using a data preprocessing technique. Preprocessing techniques include strategies to quantify the reduction of the computational cost related to pattern recognition algorithms, such as cleansing data by removing noise and inconsistent or redundant information. In addition, when considering the nature of the DBSCAN algorithm, it is noted that the computational cost will decrease if the input information is represented by a two-dimensional dataset, losing a small amount of the original information. This dataset representation can be carried out using the factor analysis dimensionality reduction algorithm [30]. The steps of dimensionality reduction using FA are presented below.

### 2.1. Data Collection Method

The data collection method for a dataset *D* considers the operation of the DBSCAN algorithm, which aims to create clusters in a two-dimensional space from said dataset *D*. As presented by Mujica et al. [31], this dataset *D* is a matrix of size m×n, where *m* is the number of row vectors $x_{i}$, i.e., experimental trials defined by a set of variables of interest in a time instant, and *n* is the number of column vectors $\nu_{j}$ of one variable of interest such as the one extracted by a network of strain sensors or accelerometers.

The number of column vectors can also be assumed as the number of dimensions of the dataset *D*, which is represented in matrix form as
(1)$$D_{n\times m}=\left[\begin{array}{ccccc} d_{11} & d_{12} & \cdots & d_{1j} & d_{1m}\\ \cdots & \cdots & \cdots & \cdots & \cdots \\ d_{i1} & d_{i2} & \cdots & d_{ij} & d_{im}\\ \cdots & \cdots & \cdots & \cdots & \cdots \\ d_{n1} & d_{n2} & \cdots & d_{nj} & d_{mn} \end{array}\right],$$

### 2.2. Data Normalization

Normalization of the original database $D_{m\times n}$ allows the algorithm to process the information using compatible magnitudes. This facilitates the correlation among variables, which improves the precision of the clustering process. Normalization was carried out using auto-scaling, which transformed each variable into an element with zero mean and unity variance as
(2)$${\bar{d}}_{ij}=\frac{d_{ij}-\mu_{vj}}{\sigma_{v_{j}}},$$
where $\sigma_{v_{j}}^{2}$ is defined as the variance of $v_{j}$, defined by:(3)$$\sigma_{v_{j}}^{2}=\frac{1}{n-1}\sum_{n}^{i=1}{\left(d_{ij}-\mu_{v_{j}}\right)}^{2},$$
where $\mu_{vj}$ is the mean of the variable of interest $v_{j}$.

### 2.3. Dimensionality Reduction Technique

The process of dimensionality reduction was performed using the linear Factor Analysis (FA) method, which has similar characteristics to principal component analysis; as FA, PCA is also a linear technique based on orthogonal projections [32]. PCA is a widely known dimensionality reduction technique that reduces the size of the dataset based on the co-variance of the original information. Nevertheless, FA reveals underlying information hidden in the original dataset using a combination of linear variables *m*, with m<p, except for an error term with a length size equal to the original dataset. The general form of the FA method is presented using the notation presented by I. T. Jollife [33]: $$\begin{array}{cc} \; & x_{1}=\lambda_{11}f_{1}+\lambda_{12}f_{2}+\cdots +\lambda_{1m}f_{m}+\epsilon_{1}\\ \; & x_{2}=\lambda_{21}f_{2}+\lambda_{22}f_{2}+\cdots +\lambda_{2m}f_{m}+\epsilon_{2}\\ \; & \vdots \\ \; & x_{p}=\lambda_{p1}f_{p}+\lambda_{p2}f_{2}+\cdots +\lambda_{pm}f_{m}+\epsilon_{p}, \end{array}$$
where *x* represents the attributes or original variables $x_{1},x_{2},\cdots ,x_{p}$, $\lambda_{jk}$ is the factor loadings: j=1,2,⋯,p, k=1,2,⋯,m, while $f_{1},f_{2},\cdots ,f_{m}$ represent the common factors and $e_{p}$ is defined as a residual error vector of specific factors. In general, a matrix representation of FA is given by:(4)X=ΛF+ϵ.

In this case, the factor loading Λ and the common factors *F* remain unknown; thus, in contrast with a standard regression model, the FA technique can lead to different solutions, which means there will not be a single solution. FA may be represented in terms of co-variances as follows:(5)$$\sum =\Lambda \Lambda^{\prime }+\Psi ,$$
where ∑ represents the covariance or correlation matrix and Ψ the covariance of the specific factors $\epsilon_{p}$ using the maximum likelihood estimation, allowing finding more precise values of Λ and Ψ.

In addition, a rotation matrix *T* can be included in order to define different solutions for FA: $\Lambda^{*}=\Lambda T$, which, after a mathematical process *T*, is included in the rotation as $\Lambda \Lambda^{\prime }$. Varimax [34], Quartimax [35], and Promax [36] are commonly used as factorial rotation methods. The pseudo-code of FA is presented in Algorithm 1.
**Algorithm 1:** Dimensionality reduction using Factor Analysis.
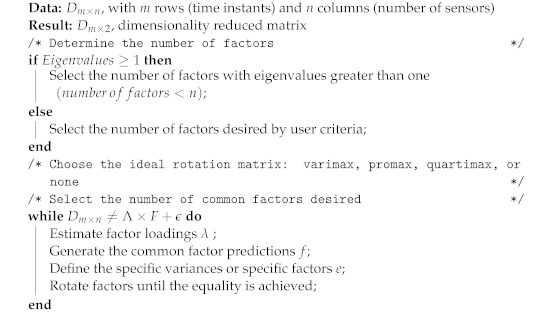


The desired rotating factors will reduce the original dimension of a dataset correlated to several specific eigenvalues that can describe the retained information. Several techniques are used to quantify the information retained after the dimensionality reduction. These techniques include the “eigenvalues greater than one” rule, the definition of the cumulative variance over 80%, and the scree-plot rule, which is a graphic method where the breaking point of a curve of factors against eigenvalues is identified as the point related to the number of appropriate eigenvalues.

In short, as common factors, $f_{m}$ will preserve hidden relationships among variables, and the DBSCAN algorithm performs its clustering process in a two-dimensional space; a strategic approach is to preserve the first two common factors from the original dataset, which can be represented and plotted in a two-dimensional space. It is also expected that different magnitudes related to specific values inside the dataset remain preserved after the dimensionality reduction process, allowing DBSCAN to identify well-defined clusters. In other words, the original dataset of dimension $D_{m\times n}$ of *m* variables and *n* sensors or dimensions can be represented in a lower dimension using the FA’s first two common factors as a $D_{m\times 2}$ dataset. Moreover, the new reduced dataset can now be graphically represented in a Cartesian coordinate system, in which every row from dataset $D_{m\times 2}$ represents an x,y point in space. An example of the dimensionality reduction of a dataset of six-column vectors containing two classes is presented in Figure 1, in which the first three dimensions are plotted. After performing a dimensionality reduction using the first two common factors, the projection of these two classes is observable. Figure 1a presents a graphic representation of an artificial dataset with $D_{1000\times 6}$. Figure 1b illustrates the projection of the first two common factors of the artificial dataset with $D_{1000\times 2}$.

## 3. Dbscan Algorithm

As previously mentioned, DBSCAN is an unsupervised density-based algorithm. It was designed to define specific clusters from a particular dataset, usually in a two-dimensional space $D_{m\times 2}$, without the need for predefined class labels. However, the algorithm is not fully automatized, thus the necessity to define two entry parameters Eps and MinPts. Nonetheless, the DBSCAN algorithm is still relevant due to its exploratory characteristics and its acceptable computational cost O(nlogn), for large datasets [37].

These initial parameters allow the algorithm to define a specific group of correlated points depending on their Euclidean distance defined by a circle with radii Eps and a minimum specified number of correlated elements MinPts. Moreover, these entry parameters define a group of points that may have no correlation with any of the discovered clusters and will be treated as noise. The pseudo-code of the DBSCAN algorithm is presented in Algorithm 2.

The DBSCAN algorithm follows a series of rules to define clusters from a specific dataset *D*, considering two arbitrary points *p* and *q* from *D*. These rules are defined as follows:Eps-neighborhood of point: The Eps-neighborhood of a point *p*, denoted by $N_{\mathrm{Eps}}\left(p\right)$, is defined by $N_{\mathrm{Eps}}\left(p\right)=\left\{q\in D|distp,q\leq \mathrm{Eps}\right\}$.Directly density-reachable: A point *p* is directly density-reachable from a point *q* if:–$p\in N_{\mathrm{Eps}}\left(q\right)$.–The core point condition is reached, i.e., $N_{\mathrm{Eps}}\left(q\right)\geq \mathrm{MinPts}$.Density-reachable: A point *p* is density-reachable from a point *q* if there is a set of points $p_{1},\ldots ,p_{n}$, with $p_{1}=q$ and $p_{n}=p$, such that $p_{i+1}$ is directly density-reachable from $p_{i}$.Density-connected: A point *p* is density-connected to a point *q* if there is a point *o* such that *p* and *q* are density-reachable from *o*.Cluster: Let *D* be a specific dataset. A cluster is a non-empty subgroup from dataset *D* that meets the following criteria:–Maximality ∀p,q: if p∈C and *q* is density-reachable from *p*, then q∈C.–Connectivity ∀p,q∈C, then *p* is density-connected to *q*.Noise: Let $C_{1},\ldots ,C_{k}$ be the clusters of dataset *D*. Noise is defined as the set of points in the dataset *D* not belonging to any cluster $C_{i}$, that is $p\in D|\forall i:p\notin C_{i}$.
**Algorithm 2:** Clustering using DBSCAN algorithm.
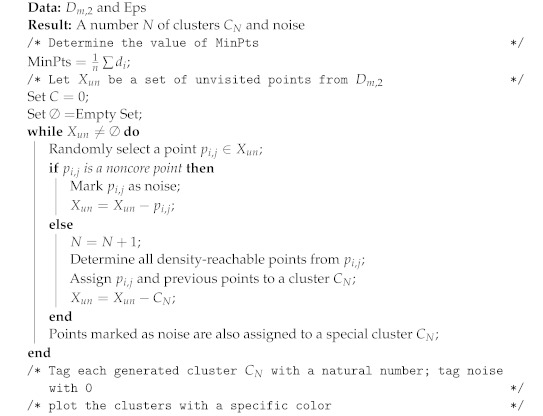


As mentioned before, the selection of the initial parameters will impact the algorithm’s overall clustering precision and computational complexity. Therefore, a strategy is needed to define the parameters MinPts and Eps seeking to remove human handling and improve the precision in the overall process. Another important characteristic of the DBSCAN algorithm is its ability to automatically define outliers as noise, excluding undesired or redundant information.

### 3.1. Definition of Parameter MinPts

The function nearest neighbor is considered to define the parameter MinPts. The function nearest neighbor defines specific distances in the proximity of an element belonging to a dataset $D_{m\times n}$ using the Euclidean distance among variables. These distances can be assumed as particular densities in the cloud of points. As presented by Gaonkar and Sawant Gaonkar and Sawant [38], the parameter MinPts can be defined using the function sample mean of the particular densities from a specific dataset $D_{m\times n}$:(6)$$\mathrm{MinPts}=\frac{1}{n}\sum_{i=1}^{n}d_{i},$$
where $d_{i}$ is every value of density assessed by the function nearest neighbor in a specific dataset $D_{m\times 2}$ with *m* number of samples.

### 3.2. Definition of Parameter Eps Using a Fitness Proportionate Selection

As presented in the previous section, the parameter Eps can be interpreted as an imaginary radius of a circumference inscribed around an arbitrary point included in a specific dataset of dimension two $D_{m\times 2}$. It is necessary to find an adequate Eps value since this parameter affects the overall clustering process of the hybrid algorithm in terms of computational cost and precision. According to the previous statement, the algorithm may lose its clustering capacity if the parameter Eps is selected arbitrarily as the user disregards the magnitude and density of the specific dataset being analyzed. As a solution, a Genetic Algorithm (GA) based on a fitness function is considered to define a particular Eps for a dataset $D_{m\times 2}$. The GA is defined using the model presented by [39].

Typical distances or densities are determined using the function nearest neighbor for a specific dataset $D_{m\times 2}$ to define the initial population of the selected GA model. These distances are defined as the standard radii from the specific dataset. The initial density population is defined in 50 elements with magnitudes between the average radius $r_{avg}$ and the maximum radius $r_{max}$. Points with coordinates belonging to the dimensionality-reduced dataset $p_{x,y}\in D_{m\times 2}$ are selected as additional alleles in the chromosome associated with the initial population to be optimized via the GA. These points are considered the center of the possible radii. Therefore, the chromosome could have the following structure presented in Table 1.

Furthermore, the fitness function ff to be optimized will have the following outline:(7)$$ff=\frac{CR\times SD}{DR},$$
where CR is considered as the coverage ratio and is calculated as follows:(8)$$CR=\frac{|S_{p_{1},r_{1}}\cup S_{p_{2},r_{2}}\cdots \cup S_{p_{n},r_{n}}|}{\left|D\right|},$$
SD being defined the Sum of Density and evaluated as follows:(9)$$SD=\sum_{i=1}^{n}\frac{|S_{p_{i},r_{i}}|}{|r_{i}^{2}|},$$
while DR is established as the Duplicate Ratio and is calculated as follows:(10)$$DR=\frac{\sum_{i=1}^{n}\left|S_{p_{i},r_{i}}\right|}{|S_{p_{1},r_{1}}\cup S_{p_{2},r_{2}}\cdots \cup S_{p_{n},r_{n}}|};$$
in general, $S_{p_{i},r_{i}}$ can defined as the chromosome of center $p_{i}$ and radius $r_{i}$.

The crossover process was performed by selecting radii from the initial population and rearranging their position while points (x,y) remained in the same initial position. These new configurations are the offspring and are defined for a tournament selection by evaluating the fitness function. The definitions of parameters Eps and MinPts are described in Algorithm 3.
**Algorithm 3:** Selection of DBSCAN parameters using a genetic algorithm.
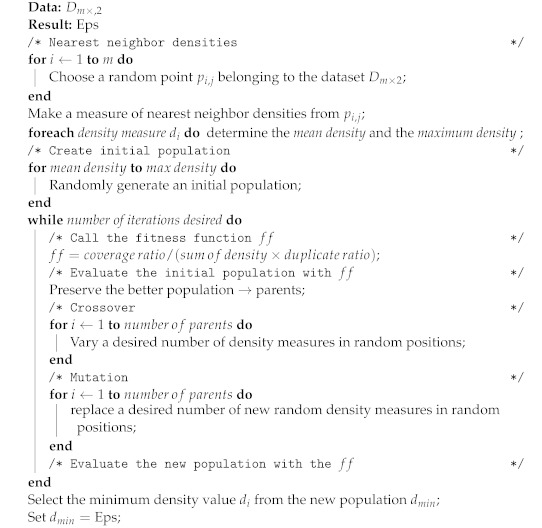


## 4. Results

### 4.1. Performance Evaluation Metrics

#### 4.1.1. Precision

The precision metric of a classifier is an evaluation parameter that is primarily used for classification performance [40]. Precision is a direct measurement of the quality of the information obtained by a clustering algorithm. For example, the precision of a classifier can be measured using the propositions presented by [41]:(11)$$\mathrm{precision}=\frac{\mathrm{tp}}{\mathrm{tp}+\mathrm{fp}},$$
where tp is defined as the true positive rate, hit rate, or recall of the clustering algorithm and is defined by:(12)$$\mathrm{tp}=\frac{\mathrm{positives}\;\mathrm{correctly}\;\mathrm{classified}}{\mathrm{total}\;\mathrm{positives}},$$
and fp is defined as the false positive rate of the clustering algorithm:(13)$$\mathrm{fp}=\frac{\mathrm{negative}\;\mathrm{incorrectly}\;\mathrm{classified}}{\mathrm{total}\;\mathrm{negatives}}.$$

#### 4.1.2. Entropy

Currently, the evaluation of Entropy has been extended as a popular metric, considering the homogeneity of a pattern recognition algorithm [42,43]. In the machine learning context, Entropy can be measured in the output parameters of the classifier as a way to define the disorder of the information processed by the algorithm. For example, the Entropy, $H\left(S\right)$, is measured as follows:(14)$$H\left(S\right)=p_{i}\log_{2}p_{i},$$
where $p_{i}$ is defined as the probability of an *i*th element belonging to a specific class.

The Entropy indicates the degree of randomness in the data set when used as a metric to estimate the data set’s uncertainty. In this regard, when applying a recognition algorithm to a specific data set, it is expected that the classified data presents a reduction in its Entropy. The Entropy difference between unclassified and classified data represents the amount of information gained after applying a classification method. This difference or information gain, $IG\left(A,S\right)$, also indicates the uncertainty reduction after splitting the data on a feature (i.e., the more significant the information gained, the greater the decrease in Entropy or uncertainty). The information gained is given as follows:(15)$$IG\left(A,S\right)=H\left(S\right)-\sum_{j=1}^{m}\frac{n_{j}}{n}\;\cdot \;H\left(S,A\right),$$
$n_{j}$ being the number of instances with a j value of an attribute A, *n* the total number of instances in the dataset, *m* the set of distinct values of an attribute *A*, $H\left(S_{j}\right)$ the Entropy of the subset of instances for attribute *A*, and $H\left(S,A\right)$ the Entropy of an attribute *A*. In the context of the DBSCAN algorithm, these futures or partitions include data in either of the following attributes: correctly classified (tp), negative incorrectly classified (fp), and noise data.

#### 4.1.3. Calinski–Harabasz Clustering Evaluation Method

Calinski and Harabasz [44] presented a clustering evaluation technique that suggests a suitable number of clusters of a specific dataset being analyzed. This exploratory technique, also named the CH Index, evaluates the cohesion or dispersion among elements considering a variance index. Following the notation of the CH Index, the technique is defined by:(16)$$CH\left(K\right)=\frac{B(K)(N-K)}{W(K)(K-1)},$$
considering B(K) as the inter-cluster covariance or divergence:(17)$$B\left(K\right)=\sum_{k=1}^{K}a_{k}\left|\right|{\bar{x}}_{k}-\bar{x}{\left|\right|}^{2};$$
furthermore, W(K) is considered as the intra-cluster covariance and is defined as:(18)$$W\left(K\right)=\sum_{k=1}^{K}\sum c\left(j\right)=k||d_{i}-{\bar{x}}_{k}{\left|\right|}^{2}$$
where *K* represents the number of clusters, $d_{i}$ is the *i*th defined cluster, and N represents the number of elements or samples.

### 4.2. Clustering Performance Analysis

The unsupervised classification methodology included the preliminary processing, processing, and postprocessing of dataset $D_{n\times m}$. The classification was performed using the MATLAB R2021b numerical programming software for Windows 11 with an Intel Core i7, 2.11 GHz processor, 8 GB of RAM, and 1 TB hard drive PC. The hybrid algorithm was created by using a combination of pre-established MATLAB functions, scripts created from the beginning, and a modified script based on the DBSCAN algorithm developed by [45]. The selection of parameters for DBSCAN was defined as proposed in Section 3.1 and Section 3.2. Finally, the FA algorithm was executed using the MATLAB built-in function Factoran, which includes the rotation method and the auto-scaling process.

To evaluate the classification performance of the hybrid algorithm FA+GA-DBSCAN, six artificial datasets were selected. The results of the different datasets classified with the FA+GA-DBSCAN algorithm are observed in Figure 2. DBSCAN parameters were defined automatically using a genetic algorithm for each dataset as presented in Section 3; however, the GA method employed a randomized basis; therefore, in order to have control over the selection of parameters Eps and MinPts, each clustering experiment was executed 30 times, and then, the standard deviation was measured for each case. The results of the obtained values for the mentioned parameters are presented in Table 2. Moreover, the well-known pattern recognition algorithm K-means was selected as a comparison benchmark. Although K-means is considered an unsupervised clustering method, it correlates elements taking into account a number of centroids selected a priori by the user. The clustering performance of K-means is illustrated in Figure 3. The results of K-means were also used for a comparative study of FA+GA-DBSCAN’s performance employing the Calinski–Harabasz clustering evaluation method. Information related to the comparative study is presented in Table 3.

A two-dimensional dataset called Half-ring (Jain) was proposed by Jain and Fred [46]; this dataset is composed of two classes with uneven densities between clusters. Each cluster is well separated, and the top one is made up of 97 elements and the bottom one of 276 elements. As presented in Figure 2a, the algorithm was able to identify two groups; however, certain elements were defined as noise, and the precision was reduced as some elements from the top cluster were placed in the group belonging to the bottom cluster.

Another dataset named Aggregation is a two-dimensional, heterogeneous synthetic distributed dataset of seven classes and 788 elements, proposed by [47]. As a result, the FA+GA-DBSCAN detected eight clusters, and some elements were considered as noise, as presented in Figure 2b; nevertheless, the precision of the algorithm was not quite affected.

The artificial dataset Compound was presented in 1971 by Zahn C. [48]. This two-dimensional dataset presents six groups with different densities and shapes and is one of the most-common datasets for clustering validation. It was evident that the hybrid algorithm was not plausible in terms of a correct grouping in most of the presented clusters, as shown in Figure 2c. Its precision was low considering that the DBSCAN parameters were defined to cover an overall density. In general, three clusters were determined. The evaluation of precision is presented in Figure 4.

MDCgen is a synthetic multidimensional dataset produced by the algorithm developed by Iglesias et al. [49]. The dataset generator is capable of producing artificial n-dimensional datasets including outliers or noise. For this study, a three-class dataset with six dimensions was generated; furthermore, it possesses 2000 observations and 100 outlier points. The dimensionality reduction process proposes that two common factors are enough to represent 77.810% of the original variability. Clustering performance is presented in Figure 2d.

The high-dimensional dataset, Dim064, reported by [50], was also considered for a clustering analysis. The relevance of this dataset relies on the need to evaluate the clustering algorithm in a high-dimensional instance. The dimensionality reduction process suggests that 15 common factors are needed to represent 99.890% of the original variability. Nevertheless, the fist two common factors were selected for the clustering process. The synthetic Gaussian clusters are well separated even for this higher-dimensional case. In terms of precision, the clustering results of FA+GA-DBSCAN were satisfactory. As presented in Figure 2e, the algorithm was able to group almost all clusters represented by the common factors from the FA dimensionality reduction process.

Finally, the multivariate dataset called Wine [51] was considered for validation purposes. This dataset presents 13 attributes, which belong to wine characteristics such as color intensity, alcohol, and minerals, among others, and three classes related to three different cultivars. A dimensionality reduction was performed considering the study of cumulative variance. As a result, three common factors are recommended to be retained, as they represent 66.530% of the original variability. Nonetheless, the first two common factors were selected for the clustering process. The algorithm’s precision was acceptable, as presented in Figure 4; the evaluated dataset was grouped into three different clusters, as indicated in Figure 2f.

As previously mentioned, the clustering performance was measured using the external clustering evaluation metrics Entropy and precision and a comparative study using the Calinski–Harabasz clustering evaluation method. As shown in Figure 4, the clustering algorithm is capable of grouping well-condensed datasets with significant precision; nevertheless, the Entropy of the resulting clusters on these types of datasets is almost invariant. On the other hand, the information gain is relatively low in condensed datasets. Furthermore, it is evident that the automatic grouping process held by FA+GA-DBSCAN can perform clustering with similar characteristics as those presented by K-means, considering the Calinski–Harabasz clustering estimation technique. However, FA+GA-DBSCAN presents an advantage when evaluating datasets with an unknown number of classes, taking into consideration that the number of centroids or classes is not previously needed; this ensures a decent level of reliability of the results in an exploratory analysis performed by the presented methodology. The Entropy and precision of K-means are also reported in Figure 5.

## 5. Case Studies

### 5.1. Aircraft Engine Degradation

The work developed by Saxena et al. [52] presented a group of datasets of an aircraft’s engine thermodynamic model simulation using the Commercial Modular Aero-Propulsion System Simulation (C-MAPSS) software. Simulations were carried out under various operational conditions with induced damage propagation. The free access engine dataset is available in the Prognostics Data Repository from the National Aeronautics and Space Administration (NASA) [53]. The study included four different output cases with different altitude, Mach number, and temperature conditions. Each output case behaves differently and includes a specific number of operational conditions and degradation, where outputs are gathered in a dataset matrix.

Considering the analysis proposed in Section 2.1, the dataset collection configuration consists of 21 sensors taking into account the low and high compressor and turbine temperature, pressure, flow speed, and fuel flow, and rows are defined as time instants. Hence, the dataset has a size of ${\mathrm{engine}}_{33991\times 21}$. The operational conditions and degradation of the engine are sorted in a way that is not directly quantifiable. The selected dataset includes six operational conditions and one fault mode belonging to the degradation of the high-pressure compressor.

The performed analysis began with a dimensional reduction using FA. Then, the factor selection was analyzed by the “eigenvalues greater than one” method; this suggests that the first two common factors represent 97.550% of the variability of the original information. Finally, the scree-plot, as presented in Figure 6, represents the eigenvalue of each factor belonging to the considered dataset; it clarifies the FA and illustrates FA’s capacity to retain a large amount of information in a lower dimension.

Furthermore, to perform this exploratory analysis, a matrix of ${\mathrm{engine}}_{33991\times 2}$ was obtained. The two first common factors were considered as points in a two-dimensional space and then automatically grouped by the hybrid algorithm. Parameters MinPts=6.480±0 and Eps=0.010±0 were defined, and the standard deviation of both parameters was measured after 30 equal runs. As a result, six well-condensed clusters were found. One of them is mainly traced out from the other five as presented in Figure 7. This may indicate a set of parameters related to engine degradation.

### 5.2. Lidar Dataset

The Lidar dataset is a free access dataset available in Matlab R2021b [54] for clustering analysis. Lidar is a mapping system that employs laser energy for high-resolution spatial sensing. This laser technology has been widely used for digital cartography, military applications, cellphones, and autonomous mobility. In this study, the linear dataset presents a spatial overview of a street with a vehicle and various objects such as trees and buildings. This dataset can be assumed as a two-dimensional top base view from a surveillance unmanned aerial vehicle; a contextualization image is presented in Figure 8.

The information from the Lidar dataset can be used for an exploratory analysis using clustering in order to identify possible objects in an unsupervised manner. In this two-dimensional spatial dataset, the space is limited to a range of 20 m × 20 m. The matrix considered for this study is a two-dimensional set of points of size ${\mathrm{lidar}}_{19070\times 2}$, and the scatter plot of this dataset is presented in Figure 9a. This set was therefore analyzed automatically by GA-DBSCAN. Parameters MinPts=0.207±0 and Eps=3.228±0.011 were defined after ten equal runs. The algorithm was capable of determining 13 different clusters, using a mean neighbor circumference value of 3.22 m approximately. This allows for the identification of elements such as the car in the center of the figure and the other one in front of it. Similarly, the algorithm was able to identify other obstacles, such as trees and borders belonging to the sidewalk. The exploratory result is presented in Figure 9b.

## 6. Conclusions

In summary, the evaluation of a clustering hybrid algorithm called FA+GA-DBSCAN was presented using Entropy and precision performance metrics and a comparative study employing the Calinski–Harabasz clustering evaluation method on different artificial datasets. This unsupervised pattern recognition algorithm was first developed to identify the operational conditions in a structure under various loads. The dimensionality reduction technique Factor Analysis described the information from the dataset as a combination of specific factors that could be clustered later using DBSCAN. However, DBSCAN on its own cannot automatically define clusters in a particular dataset as the parameters Eps and MinPts need to be selected before the recognition of patterns. A large number of variations of DBSCAN are still being proposed since the clustering algorithm operates according to the parameters Eps and MinPts. These parameters can be defined using many deterministic techniques, including density studies, genetic algorithms, and evaluations made by “hand” iterations, among others. As the algorithm is implemented using various parameter definition techniques, many variations of the clustering results are presented. The performance of FA+GA-DBSCAN clustering was defined using Entropy, precision, and a comparative study, which included the well-known clustering algorithm K-means. The hybrid algorithm automatically clustered datasets with condensed and scattered groups with notable precision; however, the information gained in this type of dataset was almost null as the variation of the Entropy did not change significantly.

## Figures and Tables

**Figure 1 entropy-24-00875-f001:**
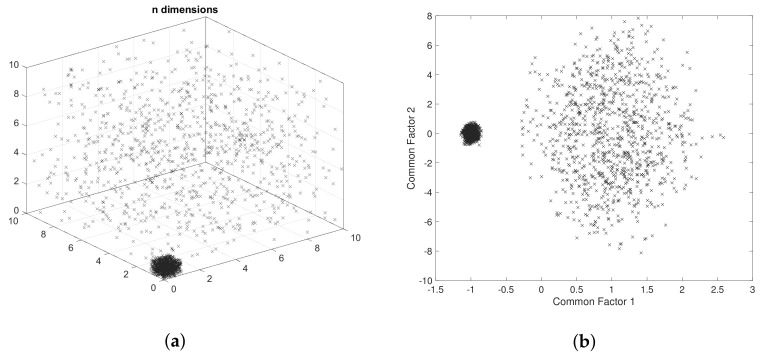
Example of a dimensionality reduction of a dataset *D*. (**a**) Three-dimensional scatter-plot of dataset $D_{1000\times 6}$. (**b**) Scatter-plot of artificial dataset projection $D_{1000\times 2}$.

**Figure 2 entropy-24-00875-f002:**
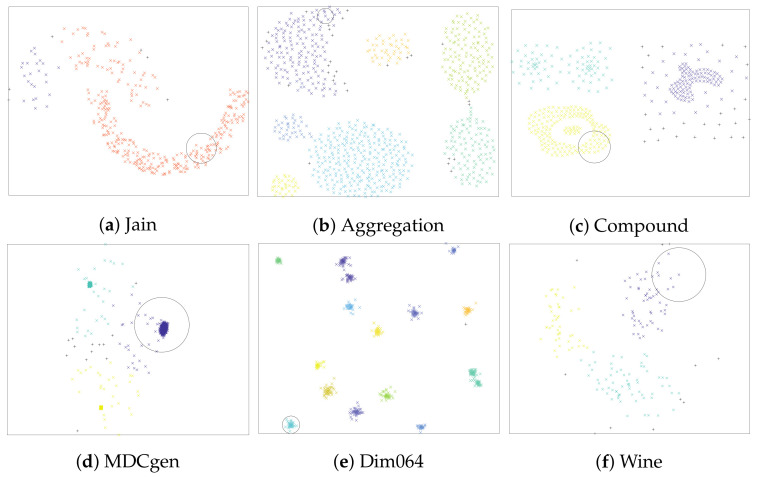
Clustering results using the FA+GA-DBSCAN algorithm with artificial datasets. Noise points are marked by “+”.

**Figure 3 entropy-24-00875-f003:**
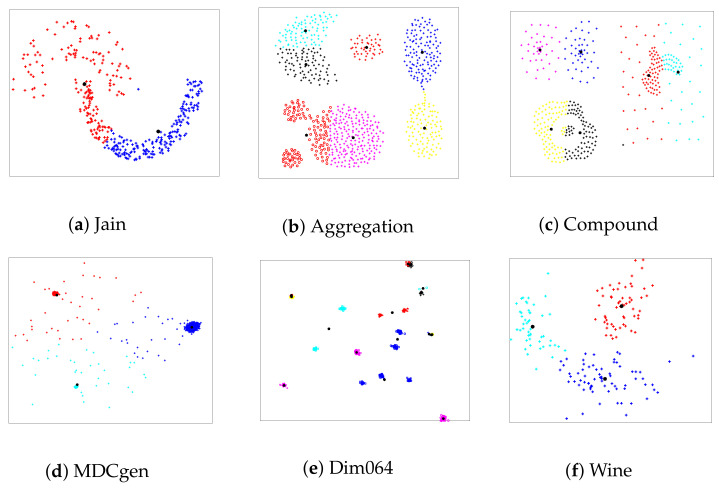
Clustering results of artificial datasets using K-means algorithm with artificial datasets.

**Figure 4 entropy-24-00875-f004:**
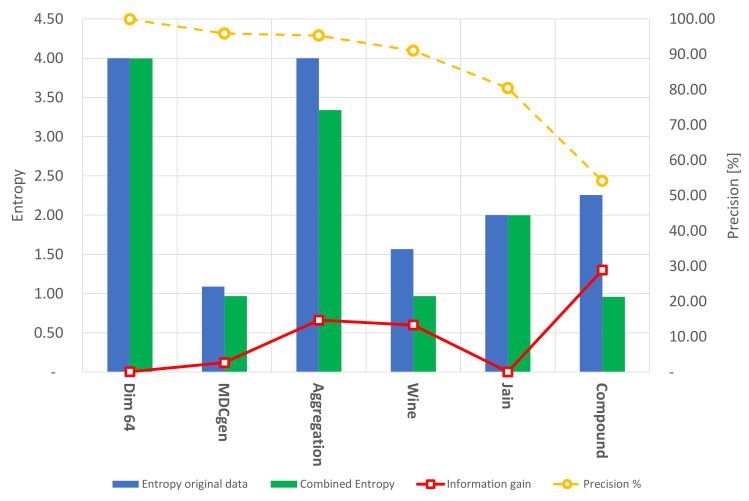
Clustering results, precision, Entropy, and information gain using the hybrid algorithm FA+GA-DBSCAN.

**Figure 5 entropy-24-00875-f005:**
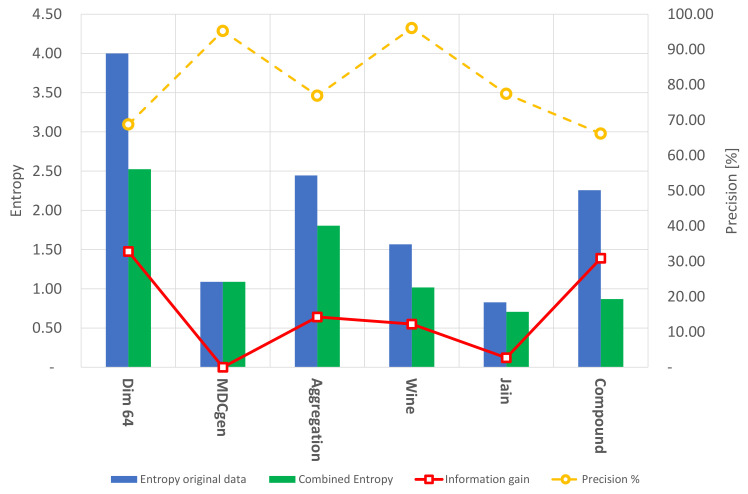
Clustering results, precision, Entropy, and information gain using K-means.

**Figure 6 entropy-24-00875-f006:**
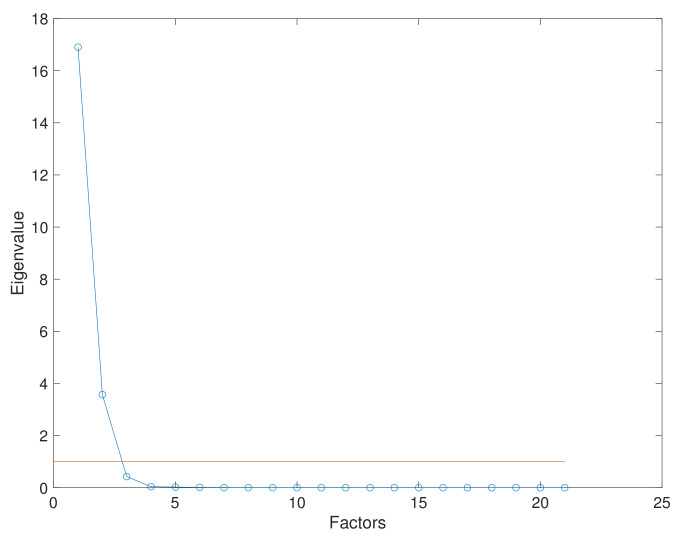
Scree-plot from the aircraft engine operational conditions and degradation dataset. Two common factors are sufficient to represent a large quantity of the original information.

**Figure 7 entropy-24-00875-f007:**
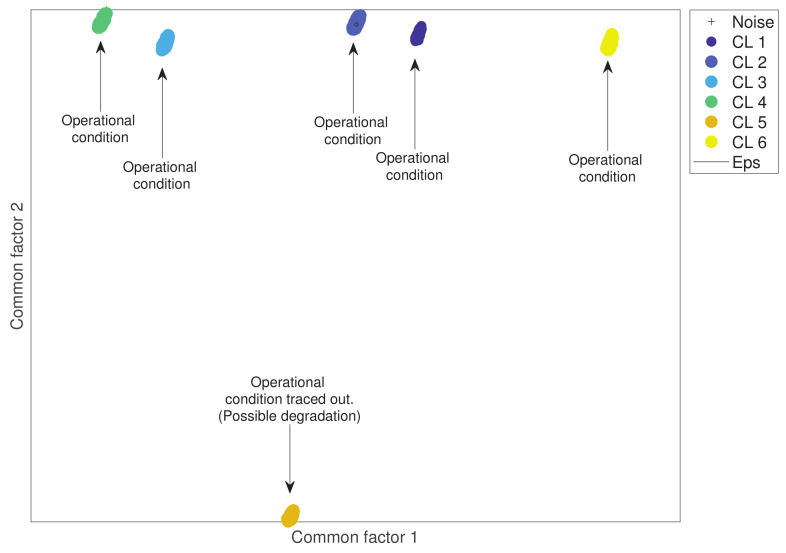
Clustering analysis of an aircraft engine considering different operational conditions and one mode of degradation.

**Figure 8 entropy-24-00875-f008:**
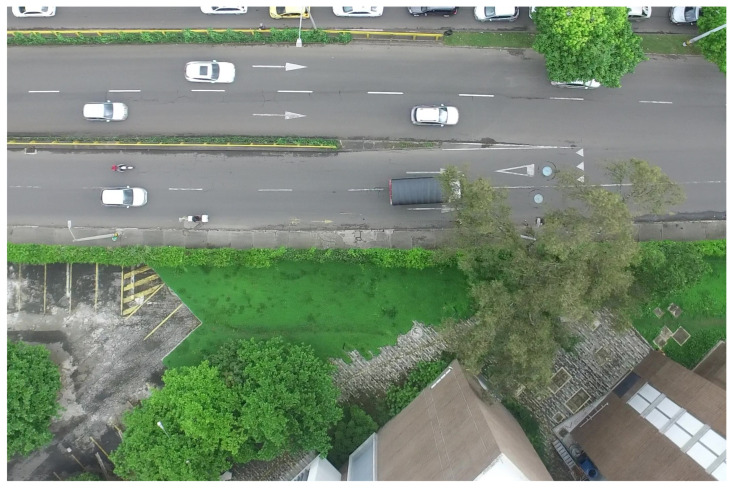
Contextualization picture considering a similar point of view to the Lidar dataset.

**Figure 9 entropy-24-00875-f009:**
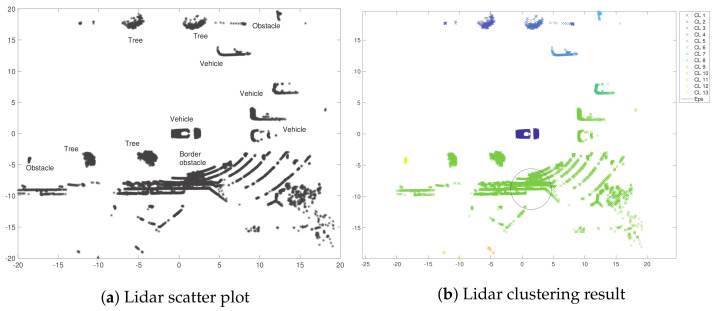
Results of the exploratory analysis using the hybrid algorithm from the Lidar dataset.

**Table 1 entropy-24-00875-t001:** A scheme of a chromosome belonging to the initial population with two alleles; one is the point *p*, and the other is the radius *r*.

Allele 1		Allele 2
*x*	*y*	Radius *r*
51.606	12.783	1.036

**Table 2 entropy-24-00875-t002:** Automatic definition of FA+GA-DBSCAN’s parameters Eps and MinPts; values are represented using their mean and standard deviation after 30 runs of the algorithm.

Dataset Name	Eps	MinPts
Aggregation	$1.130\pm 2.52\times 10^{-5}$	5.498±0.00
Compound	$2.413\pm 2.60\times 10^{-4}$	7.710±0.00
Jain	$2.550\pm 3.89\times 10^{-4}$	5.228±0.00
Dim064	$0.142\pm 1.95\times 10^{-6}$	5.898±0.00
Wine	$0.694\pm 9.90\times 10^{-6}$	23.042±0.00
MDCgen	$0.872\pm 3.71\times 10^{-5}$	26.055±0.00

**Table 3 entropy-24-00875-t003:** A comparative study of clustering performance using the Calinski–Harabasz clustering evaluation method and FA+GA-DBSCAN; C refers to cluster.

Dataset Name	Classes	Calinski–Harabasz Optimal C	C Defined by FA+GA-DBSCAN
Jain	2	9	2
Aggregation	7	6	7
Compound	2	2	3
MDCgen	3	5	3
dim064	16	16	16
Wine	3	3	3

## Data Availability

Not applicable.

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
