# Peer review of "Performance Analysis and Architecture of a Clustering Hybrid Algorithm Called FA+GA-DBSCAN Using Artificial Datasets"

_entropy, 2022, doi:10.3390/e24070875_

Round 1

Reviewer 1 Report

The main contribution of this paper is the introduction of a strategy which pairs a factorial method with density based clustering. A secondary contribution is the use of entropy to assess the clustering performance.

In my opinion, the pairing of Factor Analysis or PCA and a variant of DBSCAN should be motivated with attention. PCA and Factor Analysis are very effective when data present LINEAR correlations because only in these cases data representation in low dimensions allows to get an high percentage of explained variability. On the contrary, DBSCAN is effective when data present a non-spherical shape. Here, the hypothesis seems to be that 2 dimensions are sufficient to represent high dimensional data (after the factorial dimensionality reduction) which is a strong claim in the most of real world cases.

This contradiction emerges also in the application where four analyzed datasets are just 2-dimensional (with clusters having non spherical shape) and no preprocessing is needed, while further two datasets consider Gaussian data which are a natural choice for PCA and FA methods but don’t need the use of algorithms for non-spherical data.  It is easy to see from Fig.3 that the last two datasets don’t need density based algorithms but classic methods like k-means or EM.

Minor comments:

DBSCAN should be introduced focusing also on the kind of applications for which it is very effective. Its computational cost is HIGH (while authors claim that the computational cost is low) and this is one of the main motivations for the development of faster versions.

Line 229: genetic algorithm instead of generic

Clustering in usually evaluated through external and internal validity indexes while authors only use measures for evaluating supervised classification methods.

There are no results on the effectiveness of dimensionality reduction by PCA or FA, ie. Percentage of explained variability.

There is no comparison with other clustering methods.

Author Response

Dear Reviewer, 

Please see the attachment, 

Regards 

Reviewer 2 Report

This is an interesting paper that tackles a modification of an well known clustering algorithm, towards eliminating its dependence on external preset parameters. I appreciate the thorough review of the state of the art and the concise presentation of the methods.
The results presented by the authors are informative.

I have only one observation: the results in Table 2 should present some statistical value. To be more clear: since the proposed algorithm uses, at a certain step, a Genetic Algorithm - i think the authors should discuss a little on the output of the GA (which, definitely, provides at least slightly different results on different runs). The output of the GA further affects the functioning of the proposed algorithm.

Author Response

(The authors gave the same response as above.)

Round 2

Reviewer 1 Report

The revision improves the paper a lot. My initial concern of using methods based on linear correlations together with a clustering algorithm which provides solution to clustering problems where data present a non spherical/ellipsoidal shape, remains. However, authors provide more details than before, and results are more convincing. If other reviewers and editors find no concern on my previous consideration, I move to “accept with minor revisions”.

The further revision I ask is to specify in the introduction that DBSCAN is based on the concept of density and what this means in the field of clustering.

Author Response

(The authors gave the same response as above.)
